# Real-World Immunogenicity and Reactogenicity of Two Doses of Pfizer-BioNTech COVID-19 Vaccination in Children Aged 5–11 Years

**DOI:** 10.3390/vaccines10111954

**Published:** 2022-11-18

**Authors:** Gili Joseph, Elisheva Klein, Yaniv Lustig, Yael Weiss-Ottolenghi, Keren Asraf, Victoria Indenbaum, Sharon Amit, Or Kriger, Mayan Gilboa, Yuval Levy, Itai M. Pessach, Yitshak Kreiss, Gili Regev-Yochay, Michal Stein

**Affiliations:** 1The Sheba Pandemic Preparedness Research Institute (SPRI), and Infection Prevention & Control Unit, Sheba Medical Center, Tel Hashomer, Ramat Gan 52621, Israel; 2Sackler School of Medicine, Tel-Aviv University, Tel Aviv 69978, Israel; 3Central Virology Laboratory, Public Health Services, Ministry of Health, Tel-Hashomer, Ramat Gan 52621, Israel; 4The Dworman Automated-Mega Laboratory, Sheba Medical Center, Tel-Hashomer, Ramat Gan 52621, Israel; 5Clinical Microbiology, Sheba Medical Center, Tel Hashomer, Ramat Gan 52621, Israel; 6General Management, Sheba Medical Center, Tel Hashomer, Ramat Gan 52621, Israel; 7Pediatric Infectious Diseases Unit, Sheba Medical Center, Tel Hashomer, Ramat Gan 52621, Israel

**Keywords:** COVID-19, antibodies, children, vaccines, immunogenicity, reactogenicity

## Abstract

There are limited data concerning the immunogenicity and reactogenicity of COVID-19 vaccines in children. A total of 110 children, 5–11 years old were vaccinated with two doses (with a 3-week interval between doses) of the Pfizer-BioNTech COVID-19 vaccine and were followed for 21, 90, and 180 days after vaccination for immunogenicity, adverse events, and breakthrough infections. Ninety days after the first vaccine dose, the GeoMean (CI 95%) of IgG ascended to 1291.0 BAU (929.6–1790.2) for uninfected children and 1670.0 BAU (1131.0–2466.0) for Infected children. One hundred and eighty days after receiving the first dose of the vaccine, the titers decreased to 535.5 BAU (288.4–993.6) for the uninfected children, while only a small decline was detected among infected children—1479.0 (878.2–2490.0). The neutralizing antibodies titer almost did not change over time in the uninfected children, and even elevated for the infected children. Of the 110 vaccinated children, 75.5% were infected, with only mild COVID-19 infection symptoms. Child vaccination was found to be safe, with mild, mostly local, and of short duration, reported AEs. No serious adverse events (SAEs) were reported after vaccination. The durability of two doses of vaccine in children is longer, thus a booster may not be needed as early as in adults.

## 1. Introduction

COVID-19, caused by the SARS-CoV-2 virus, which emerged in December 2019, has brought death and serious illness to millions around the world [1].

Although adults possess a greater risk for severe infection, children can also be infected and reports of hospitalization in intensive care among children were as high as 4% during the first pandemic year [2,3,4,5,6], and the estimated risk for moderate/severe or critical acute COVID-19 was 1:3000 in children between the ages of 5 to 11. The mortality rates of children were much lower than adults and stood at 0.0008% compared to 0.05% in adults [2,3,4,5,6]. The Israeli Ministry of Health (IMOH) reported that the rate of hospitalization due to moderate-to-critical COVID-19 was 34.6–59.9/100,000 during pre-Omicron waves and 135.22/100,000 during the Omicron BA1,2 wave. Mortality was 0.2–3.6/100,000 and 2.8/100,000, respectively [7]. 

Most SARS-CoV-2 infected children are asymptomatic or have mild symptoms, and the vast majority recover without sequelae. COVID-19 symptoms reported in children are similar to those reported in adults, but usually in a milder form, and include fever, headache, fatigue, cough, nasal congestion, muscular pain, nausea, abdominal pain, diarrhea, skin rashes, ageusia, and anosmia [6,8,9,10,11,12,13,14]. 

Children at risk for severe disease are those with underlying diseases, such as neurological diseases, congenital syndromes, obesity, diabetes, hematologic diseases, malignancies, and immunodeficiency [6,15,16]. 

However, the risk of severe COVID-19 in children without underlying diseases is not negligible [6,17]: previously healthy children can suffer from symptoms such as Multisystem inflammatory syndrome in children (MIS-C), moderate to severe COVID-19, long COVID and other medical complications, as well as emotional distress in varying degrees due to the policy of lockdown, school closure, and isolation [6,9]. 

In addition, children can serve as vectors and reservoirs of the virus and thereby increase the transmission of the infection to the general population, thus endangering the population at risk [18,19,20].

For those reasons the American center for diseases control and prevention (CDC) recommended vaccination for children aged 12–17 and subsequently extended this recommendation to younger children aged 5–11 years old [21].

Phase 2,3 of the clinical study of Pfizer-BioNTech BNT162b2 vaccine of children aged 5–11 years [22] compared 1518 children who received the vaccine at a dose of 10 µg, which is one-third of the dose given to children aged ≥12 years, in a two-dose schedule, with a 3-week interval between doses, and 750 children who received a placebo. The level of neutralizing antibodies one month after the second dose of the vaccine was similar to that achieved in individuals aged 16–25-years, despite the reduced dose. The vaccine efficacy for the prevention of symptomatic disease was 90.7% in the pre-Omicron era.

Following regulatory authorization in the USA [23], the recommendations of the CDC (17), and the professional societies in Israel (6), the vaccine was approved for administration for children aged 5–11 years in Israel by the Israeli Ministry of Health (IMOH) in November 2021.

The Omicron variant was first reported in South Africa in November 2021 and was found to have a higher secondary attack rate compared with Delta [24,25], a higher rate of re-infection [26,27], and higher evasion from neutralizing activity in convalescent as well as post-vaccination sera [28,29,30,31]. 

The information in the literature regarding the immunogenicity of the vaccine among children aged 5–11 years following infection is limited to the clinical studies carried out in the era of the Delta variant [32,33], and to a single observational study, which also included information about the Omicron period [34].

More information is needed, especially given the currently dominant Omicron variant. 

In this study, we aim to examine the immunogenicity and reactogenicity of Pfizer-BioNTech BNT162b2 vaccines in children in a real-world setting during the Omicron variant surge in Israel. 

## 2. Materials and Methods

### 2.1. Cohort and Study Design

This is a prospective cohort study carried out between November 2021 and May 2022, immediately after the Delta surge in Israel (25 July 2021 to 15 November 2021), and during the Omicron surge (15 December 2021 to 15 May 2022). Healthcare workers at the Sheba Medical Center who had children aged 5–11 were approached and offered to join the study. All children who agreed to participate and had an informed consent signed by their parents were enrolled. On enrollment (day 0), they were tested for SARS-CoV-2 by a nasopharyngeal swab, and blood was drawn for serology (Table 1). The day of the first vaccine dose was defined as day 0 of follow-up (visit 1). On Day 21 (visit 2), the children received the second vaccine dose. Follow-up until day 180 included blood samples and data collections on day 21 (visit 2), day 90 (visit 3), and day 180 (visit 4). IgG and neutralizing antibody titers were determined from the blood samples collected on each visit. A follow-up online questionnaire was carried out to collect adverse events and reactogenicity of the vaccine (on the 1st and 2nd visits) (Appendix A). On each visit nasopharyngeal swabs for SARS-CoV-2 RT-PCR COVID-19 were performed and SARS-CoV-2 infections were determined, either by positive SARS-CoV-2 PCR tests performed on-site or by parent report of a positive SARS-CoV-2 test (PCR or rapid Ag test) since the last visit. Additionally, at the end of the study, parents were reached out to again to identify COVID-19 infection history. Parents of COVID-19-infected children were requested to fill out an online/telephone questionnaire regarding symptoms, and severity of the disease as well as long COVID-19 symptoms (Table 1, Appendix A). 

### 2.2. Inclusion and Exclusion Criteria

Children (male or female) between 5 and 11 years of age that agreed to participate and their parents signed a written informed consent, with no history of SARS-CoV-2 in the previous 2 weeks, and have nasopharyngeal and blood sample obtained.

History of medical conditions or chronic disease was not considered an exclusion criterion. Subjects under the age of 5 and above the age of 11 years and either blood or nasopharyngeal samples were not available for any reason were excluded.

### 2.3. Ethical Statement

The protocols (numbers: SMC-8934-21, SMC–Helsinki committee application number), were approved by the institutional review board of the Sheba Medical Center. Written informed consent was obtained from all participants.

### 2.4. Vaccines

Two Pfizer-BioNTech BNT162b2 COVID-19 vaccine doses (10 µg BNT162b2) were given three weeks apart. 

### 2.5. Immunogenicity

#### 2.5.1. SARS-CoV-2 IgG II Quant (Abbott, IL, USA)

Samples were centrifuged at room temperature, at 4000× *g*, for 4 min. Serum was tested for IgG antibodies against the SARS-COV-2 spike RBD using the commercial automatic chemiluminescent microparticle immunoassay (CMIA) SARS-CoV-2 IgG II Quant (Abbott, IL, USA) according to the manufacturer’s instructions. All IgG Antibody levels were presented in binding antibody units (BAU) per World Health Organization standard measurements.

#### 2.5.2. SARS-CoV-2 Pseudo Virus (psSARS-2) Neutralization Assay

SARS-CoV-2 Pseudo-virus (psSARS-2) Neutralization Assay was performed as previously described [35]. Following titration, 100 focus forming units (ffu) of the Wuhan psSARS-2 were incubated with two-fold serial dilution of heat-inactivated (56 °C for 30 min) tested sera. After incubation for 60 min at 37 °C, the virus/serum mixture was transferred to Vero E6 cells that had been grown to confluence in 96-well plates and incubated for 90 min at 37 °C. After the addition of 1% methyl cellulose in Dulbecco’s modified eagle’s medium (DMEM) with 2% of fetal bovine serum (FBS), plates were incubated for 24 h and 50% plaque reduction titer was calculated by counting green fluorescent foci. Sera not capable of reducing viral replication by 50% at 1 to 16 dilutions or below were considered non-neutralizing. 

### 2.6. PCR Testing

Nasopharyngeal swabs were placed in 3 mL of universal transport medium (UTM) or viral transport medium (VTM). Tests were performed according to the manufacturers’ instructions on various platforms: Allplex™ 2019-nCoV (Seegene, S., Seoul, Republic of Korea).

### 2.7. Statistical Methods 

Exposures were log-transformed, using base 10 for IgG antibody titers and base 2 for neutralizing antibody titers. The distribution of variables among the study population was described as appropriate for the variable types in the entire study population and stratified by eventual infection.

For the crude analysis, the GMT and 95% confidence intervals of each serology test were compared to the baseline and between those infected and uninfected.

## 3. Results

During two weeks in November 2021, 120 children, aged 5–11 years old with no history of previous SARS-CoV-2 infection were recruited. Of them, 10 children were found positive between day 0 and day 21 and were excluded from the analysis. 

The average age of the children was 9.32 (±1.94) years (median 9.39, range 5–11). According to parental reports, except for one child who has hyperactive airway disease treated with inhalers, none of the children included in our cohorts had a chronic illness. Four of the children receive regular treatment for attention deficit hyperactivity disorder (ADHD). 

During the follow-up period, within the Omicron BA.1, BA.2 surge in Israel, 83 out of 110 (75.5%) vaccinated children (with 2 doses), had documented breakthrough infections (Table 2). 

### 3.1. Immunogenicity

At baseline (day 0), the GMT (CI 95%) of IgG antibodies against the SARS-COV-2 spike RBD was 0.44 BAU (0.36–0.55), (Figure 1A and Appendix A). At day 21 (before receiving the second vaccine dose), the IgG titer increased to 178.5 BAU (129.9–245.3). At this time point, all 110 included children were negative for SARS-CoV-2.

Ninety days after the first dose (visit 3), the GMT (CI 95%) of IgG antibodies against the SARS-COV-2 spike RBD ascended to 1523.0 BAU (1163.0–1996.0). However, by this time, 83 children were infected. The GMT was 1291.0 BAU (95% CI 929.6–1790.2) for the uninfected children and 1670.0 BAU (1131.0–2466.0) amongst the infected children (*p* > 0.05). The GMT (CI 95%) of IgG antibodies 180 days after receiving the 1st vaccine dose decreased to 535.5 BAU (288.4–993.6) in the uninfected children, while only a small decline was detected among infected children–1479.0 BAU (878.2–2490.0) (* *p* < 0.05). 

Figure 1B (and Appendix A) shows the neutralizing antibodies titers during the study. At baseline (day 0), the GMT (CI 95%) was a titer of 0.10 (0.10–0.12). At day 21 the titer increased to 78.8 (54.4–114.3). By 90 days after receiving the first vaccine dose, neutralizing antibody GMT titers increased to 664.0 (425.6–1036.0) in the uninfected group and 1057.0 (645.9–1730.0) in the infected group (*p* > 0.05). At the end of the study, 180 days after receiving the first vaccine dose, the neutralizing antibody GMT titer did not wane and the GMT in the uninfected group was 658.8 (291.9–1487.0). Among the infected group, a continued increase in neutralizing antibody titers was observed to 2250.0 (1185.0–4272.0) (* *p* < 0.05). 

### 3.2. Safety

Local pain at the injection site was common, but other local or systemic symptoms were rare (Table 3), and no SAEs were reported. 

In total, 83 (75%) children reported any reactogenic response. Of these, most had local symptoms at the injection site: 55 children had local pain after vaccination. Five children (5.7%) had redness at the injection site as well as swelling. Only one child had itching symptoms at the injection site after vaccination. All local symptoms resolved within two days. The systemic AEs reported were: fever over 37.5 °C, fatigue, myalgia, and headache. No other systemic symptoms were reported and none of the children were hospitalized (Table 3).

### 3.3. Breakthrough Infections

During the study period, the Omicron surge took place in Israel (BA.1 and BA.2) [36]. Of the 110 vaccinated children, 83 (75.5%) had a documented infection, mainly in the months of January to March 2022 (Figure 2). Of these, 37.4% had asymptomatic infection detected by PCR or Ag testing. All symptomatic infections were mild and lasted for an average of 3.5 (+/−0.71) days. Only 28.9% of the infected children were reported to have a fever. The most common COVID-19 symptoms were fatigue/weakness (44.6%), rhinorrhea (39.8%), headache ((34.9%), and sore throat (36.7%). There were no hospitalizations, and only three children (3.6%) experienced long COVID symptoms (Table 4 and Appendix A). 

## 4. Discussion

In this study we show that children that received two doses of the Pfizer-BioNTech BNT162b2 vaccine developed a substantial and durable immune response, the rate of systemic adverse events was low among these children, and, though a high rate of children developed COVID-19 infection symptoms, they were universally mild. 

Following a successful phase 2,3 study [22], the Pfizer-BioNTech BNT162b2 vaccine was approved for children aged 5–11. An initial clinical trial reported no SAEs and good immunogenicity with a dose that was three-fold reduced compared to that distributed to adults. Accordingly, global health authorities recommended vaccination for 5–11 year-old children taking into account that the high burden of COVID-19 disease in children and the COVID-19 vaccine efficacy outweigh the possible risks of vaccination [6,17]. Clinical studies [22], international [37,38], and national [7] Israeli MOH data, reported isolated cases of SAEs, without cases of myocarditis in this age group. Our findings are consistent with the above, only mild, mostly local, and of short duration, AEs were reported and no SAE was reported, though this cohort is not powered to detect SAE. 

As far as we know, this is the first real-world prospective longitudinal study that followed immunogenicity in healthy 5–11 year-old children. There are a few other real-world studies that examined immunogenicity, but they are among adolescents, among children with chronic conditions, including immunodeficiency, and overall, they demonstrated good immunogenicity, except for during chemotherapy treatment [15,39,40,41]. 

The waning of vaccine effectiveness against SARS-CoV-2 infection of the first two vaccine doses has been well described in many adult studies [42,43,44,45], including at six months after receipt of the second dose of the Pfizer-BioNTech BNT162b2 vaccine, especially among men, persons 65 years of age or older, and persons with immunosuppression [46]. Here, we report a very slow waning of IgG titers, much slower than reported in adults [46], and no waning of neutralizing antibody titers within 180 days. Moreover, among infected children, no waning was observed. 

This study was carried out during the Omicron wave, which led to high rates of breakthrough infections, as reported in adults, and adolescents [47,48]. We report that ~75% of the children in our cohort were infected with the Omicron variant, all of them mildly, with no hospitalization. The Israeli MOH [49], reported only ~37% of children aged 5–11 years old diagnosed with SARS-CoV-2 infection during this period. The MOH report probably does not reflect the true incidence in children, because many children did not verify infection with SARS-CoV-2 testing and many asymptomatic infections occurred. In our study, we observed higher rates of infection due to active surveillance during the study visits. Moreover, the study was carried out among children of healthcare workers that have better compliance for SARS-CoV-2 diagnostic tests. We cannot be sure that the mild clinical features of reported infections are a result of vaccination or simply due to a milder form of pediatric age.

Although the design of our study does not allow us to estimate vaccine effectiveness against infection, symptomatic disease, or hospitalization, our observation of a high incidence of Omicron breakthrough infections is consistent with the literature, regarding the high vaccine escape of the Omicron variant and its sub-lineages (BA.1, BA.2, BA.2.12.1, BA.4, and BA.5). 

As with adolescents and adults [50,51,52], for 5–11 year-old children, effectiveness in preventing infection has been reported to be attenuated; two doses of Pfizer-BioNTech BNT162b2 vaccine reduced the risk of Omicron infection by 31% [53]. A large-scale multicenter study from the US demonstrated moderate effectiveness for symptomatic disease that decreased rapidly: vaccine effectiveness against symptomatic infection for children 5 to 11 years of age was 60.1% 2 to 4 weeks after dose two and 28.9% during month 2 after dose two [54]. An additional study demonstrated 48% effectiveness (95% CI, 29 to 63) at 7 to 21 days after the second dose, with a trend toward higher vaccine effectiveness in the youngest age group (5 or 6 years of age) than in the oldest age group (10 or 11 years of age) [55]. The effectiveness against laboratory-confirmed COVID-19-associated hospitalization among children aged 5–11 years vaccinated with two doses was 74–86% [56,57,58].

Despite the high incidence of infections in our study, we report, no significant symptoms, moderate or severe disease, and only three cases of persistent mild COVID symptoms (more than 2 weeks after the initial illness). 

It is noteworthy that Israeli MOH data [49], recorded a total of 382 (rate of 30/10^5^) COVID-19 –related hospitalizations among Israeli children aged 5–11 years during our study period (the Omicron surge). Of them, 40 (rate of 3.2/10^5^) were in a moderate/severe/critical condition and 3 died. However, vaccination data was unavailable.

The high incidence of infections occurred despite a very slow waning of IgG antibodies and no waning of neutralizing antibodies. This highlights the high immune escape of Omicron VOCs. A booster dose will probably further increase the antibody titers which may increase vaccine effectiveness. However, this is yet to be shown in children. 

There are some limitations to the study: the design of our current study (with no unvaccinated children examined) does not allow the assessment of vaccine effectiveness and the sample size is too small for detecting rare SAEs. There is a relatively large number of children lost to follow-up, yet 29 children had a full 180 days of follow-up of antibody response. It is highly challenging to convince parents of healthy children in this age group to enroll in a study with blood sampling. 

## 5. Conclusions

We found that child vaccination is safe, with mild, mostly local, and of short duration, reported AEs. Within the 180 days’ follow-up, the waning of Ab titers was slow and negligible as opposed to that previously described in adults. Thus, the durability of two vaccine doses in children is longer. Despite this, Omicron incidence was high and further studies are required to show if a booster dose, which is expected to further increase the antibody titers, will increase vaccine effectiveness against Omicron VOCs infection and disease. Additionally, future research is needed to determine the incidence of severe infections in healthy children vaccinated with two doses in order to define the need for a booster dose and adaptation for VOCs in such populations.

## Figures and Tables

**Figure 1 vaccines-10-01954-f001:**
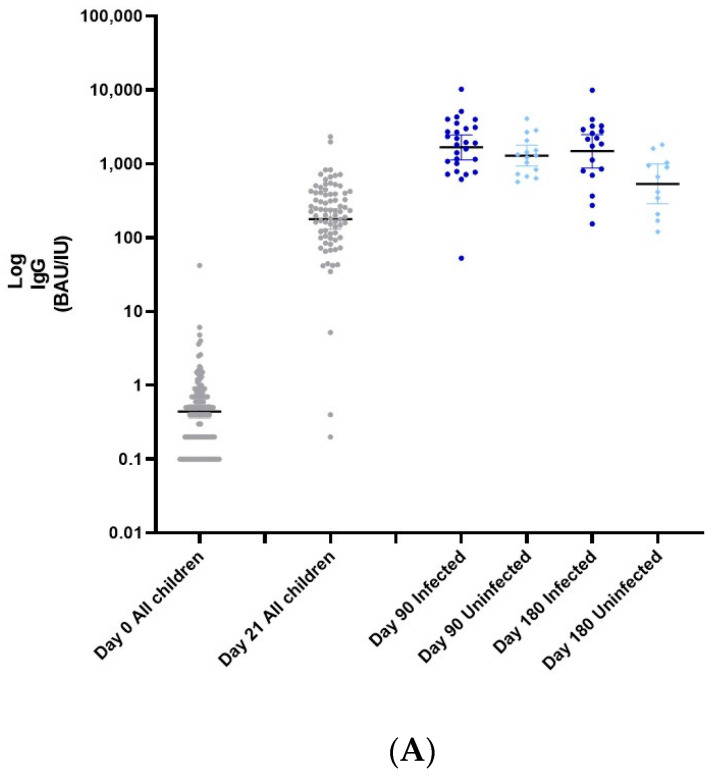
(**A**) GeoMean (CI 95%) of IgG antibodies against the SARS-CoV-2 spike RBD titer of infected and uninfected children at day 0, 21, 90, and 180. (**B**) GeoMean (CI 95%) of neutralizing Ab of Infected and uninfected children at day 0, 21, 90, and 180.

**Figure 2 vaccines-10-01954-f002:**
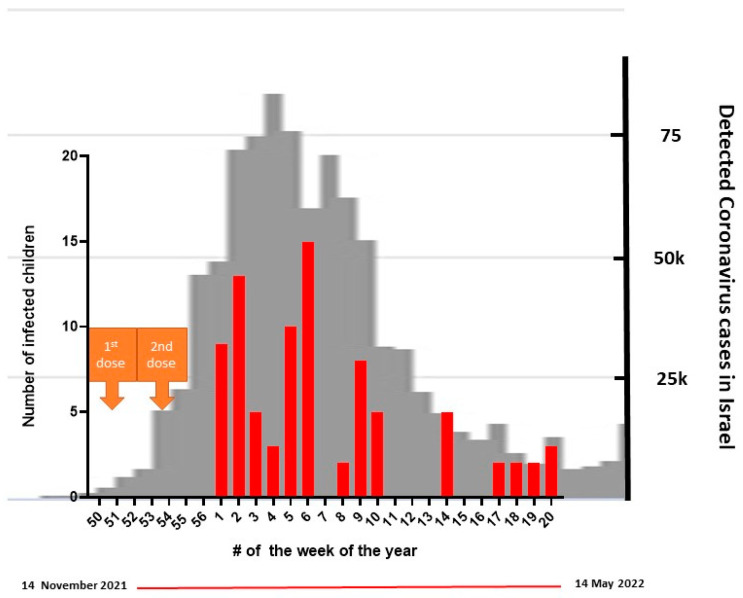
Breakthrough infections relative to the national BA.1/BA.2 wave. The grey background describes the BA.1/BA.2 national wave. The red bars represent the number of children infected in our cohort during the Omicron wave.

**Table 1 vaccines-10-01954-t001:** Study follow-up schedule and visit compliance.

Visit	1	2	3	4
Day	Day 0	Day 21 ± 4	Day 90 ± 7	Day 180 ± 7
Vaccine Dose	1st	2nd		
Number of children tested *	110	86	45	29
Number questionnaires responses	53	30	110	110
Number of infected children **	0	0	70	83

* IgG and neutralizing Ab testing ** by the time of the visit as determined by either PCR/antigen test or parents’ questionnaire.

**Table 2 vaccines-10-01954-t002:** Study population.

	Total	Infected	Uninfected
N (%)	110 (100%)	83 (75.5%)	27 (24.5%)
Age (years)			
Mean (SD)	9.18 (1.97)	9.32 (1.94)	8.65 (2.04)
Median (Range)	9.39 (5–11)	9.86 (5–11)	8.53 (5–11)
Gender = female (%)	41 (37.3%)	32 (38.6%)	9 (33.3%)

**Table 3 vaccines-10-01954-t003:** Adverse events of first and second dose of BNT126b2 vaccine.

Symptoms	First Dose of BNT126b2 Vaccine N = 52		Second Dose of BNT126b2 Vaccine N = 31	
	N (%)	Symptom Duration (Days) Mean (SD)	N (%)	Symptom Duration (Days) Mean (SD), Median
Any local symptom	37 (71.1%)		20 (64.5%)	
Pain at the injection site	37 (71.1%)	2.07 (0.94)	18 (58.1%)	2.40 (1.10), 2
Erythema	3 (5.7%)	2.07 (0.94)	2 (6.5%)	2.40 (1.10), 2
Edema	3 (5.7%)	2.07 (0.94)	0 (0%)	
Itching at the injection site	1 (1.9%)	2.07 (0.94)	1 (3.2%)	2.40 (1.10), 2
Other local symptoms	1 (1.9%) (abdominal pain)		0 (0%)	
Any systemic symptom	12 (23.1%)		7 (22.6%)	
Fever over 37.5 °C	2 (3.8%)		4 (12.8%)	1.6 (1.1), 1.5
Fatigue and weakness	9 (17.0%)	5.3 (4.4), 5	5 (16.0%)	3.4 (2.1), 3
Myalgia	6 (11.4%)	3.3 (2.3), 3	2 (6.4%)	5.5 (0.7), 5.5
Lymphadenopathy	5 (9.2%)	3 (0%), 3	2 (6.4%)	1
Headache	7 (13.2%)	3.3 (1.7), 3	4 (12.8%)	2.5 (1.9), 2.0
Facial nerve palsy	0 (0%)		0 (0%)	
Paresthesia	0 (0%)		0 (0%)	
Systemic Allergic reaction	0 (0%)		0 (0%)	
At least one day of school absence	8 (15.1%)	1.63 (1.41), 1	7 (21%)	1.2 (0.44), 1
Medical treatment	1 (1.9%) Visit a pediatrician		2 (6.4%) Analgesics, antibiotics	
Hospitalization	0 (0%)		0 (0%)	

**Table 4 vaccines-10-01954-t004:** Breakthrough COVID-19 symptoms.

	Symptom	N (%)
1.	any symptoms	52 (62.6%)
2.	fatigue or weakness	37 (44.6%)
3.	rhinorrhea	33 (39.8%)
4.	headache	29 (34.9%)
5.	sore throat	28 (33.7%)
6.	fever above 37.5 for up to 2 days	24 (28.9%)
7.	cough	20 (24.1%)
8.	Gastrointestinal symptoms	8 (9.6%)
9.	fever above 37.5 for more than 2 days	7 (8.4%)
10.	anosmia or ageusia	2 (2.4%)
11.	shortness of breath	1 (1.2%)
12.	hospitalization	0 (0%)
13.	any other symptoms	2 (2.4%) (myalgia, hallucinations)
14.	symptomatic illness length (days) COVID-19 Mean (SD), Median	3.5 (0.71), 3.5
15.	school absence (days) Mean (SD), Median	5.72 (1.48), 5
16.	persistent COVID symptoms (more than 2 weeks after the initial illness)	3 (3.6%)

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
