# Peer review of "Real-World Immunogenicity and Reactogenicity of Two Doses of Pfizer-BioNTech COVID-19 Vaccination in Children Aged 5–11 Years"

_vaccines, 2022, doi:10.3390/vaccines10111954_

Round 1

Reviewer 1 Report

Estimated Authors,

I'm gratulating with you for this concise, well written and highly significant paper on the use of SARS-CoV-2 vaccines in pediatric age individuals in Israel during the SARS-CoV-2 Omicron wave.

Even though the study design is associated with some limits that have been properly addressed by Authors in their discussion, what we can grasp from this study is that a) vaccines have no significant side effects even in children, b) a sustained immune response has been achieved, but c) breakthrough infections still occur even with high GMC levels of IgG (as in the adults, indeed).

In fact, I've no request for amendments, and only some minor suggestions for your discussion, as follows:

1) Authors should stress that studying the effect of SARS-CoV-2 in children from a clinical point of view is substantially complicated by the usually mild clinical features of SARS-CoV-2 infection in Children. In other words, but Authors have hinted to this point, we cannot be sure that the mild clinical features of reported infections may be a consequence of the vaccination or simply from the pediatric age infection;

2) Authors should discuss whether having included in their cohort children of HCWs may have increased the risk for pediatric age infections compared to the general population (i.e. are these children at higher risk to be infected in their household compared to other children?)

3) could being the child of a HCW increased the reporting rate of symptoms? In other words, could be the still satisfying report of adverse effect somehow been inflated by the higher sensitivity of their parent to clinical symptoms?

4) a minor typo in row 105: "SARS-CoV-2 RT-PCRCOVID-19" instead of "SARS-CoV-2 RT-PCR COVID-19"

Well done

Author Response

We thank you for the opportunity to revise our manuscript. Here, we address each and every point raised by the reviewers:

Reviewer 1

We thank you for your prompt and constructive review. Below are the answers relating to all the comments:

I'm gratulating with you for this concise, well written and highly significant paper on the use of SARS-CoV-2 vaccines in pediatric age individuals in Israel during the SARS-CoV-2 Omicron wave.

Even though the study design is associated with some limits that have been properly addressed by Authors in their discussion, what we can grasp from this study is that a) vaccines have no significant side effects even in children, b) a sustained immune response has been achieved, but c) breakthrough infections still occur even with high GMC levels of IgG (as in the adults, indeed).

Answer- We thank the reviewer for the very positive review.

In fact, I've no request for amendments, and only some minor suggestions for your discussion, as follows:

  • Authors should stress that studying the effect of SARS-CoV-2 in children from a clinical point of view is substantially complicated by the usually mild clinical features of SARS-CoV-2 infection in Children. In other words, but Authors have hinted to this point, we cannot be sure that the mild clinical features of reported infections may be a consequence of the vaccination or simply from the pediatric age infection;

Answer- We agree with the reviewers' comment, thus added a sentence in the discussion (line 258): “We cannot be sure that the mild clinical features of reported infection is a result of vaccination or simply due to a milder form of pediatric age. "

  • Authors should discuss whether having included in their cohort children of HCWs may have increased the risk for pediatric age infections compared to the general population (i.e. are these children at higher risk to be infected in their household compared to other children?)

Answer- Current literature does not show that HCW are at increased risk at work. In fact, HCW COVID-19 infections are mostly acquired in the community, outside of the Health Care facility (see references below) . Therefore, our explanation for the higher incidence in HCW's children compared to MOH data is as explained:

" Moreover, the study was carried out among children of healthcare workers that have better compliance for SARS-CoV-2 diagnostic tests."

         Gholami M, Fawad I, Shadan S, Rowaiee R, Ghanem H, Hassan Khamis A, et al. COVID-19 and healthcare workers: A systematic review and meta-analysis. Int J Infect Dis. 2021 Mar; 104:335–346.

         Jacob JT, Baker JM, Fridkin SK, Lopman BA, Steinberg JP, Christenson RH, et al. Risk Factors Associated With SARS-CoV-2 Seropositivity Among US Health Care Personnel. JAMA Netw Open. 2021 Mar 1;4(3): e211283.

  • could being the child of a HCW increased the reporting rate of symptoms? In other words, could be the still satisfying report of adverse effect somehow been inflated by the higher sensitivity of their parent to clinical symptoms?

Answer-We totally agree with the reviewers' comment and mentioned it in the discussion (Discussion- paragraph 4, last sentence (line 257): " Moreover, the study was carried out among children of healthcare workers that have better compliance for SARS-CoV-2 diagnostic tests. "

  • a minor typo in row 105: "SARS-CoV-2 RT-PCRCOVID-19" instead of "SARS-CoV-2 RT-PCR COVID-19"

Answer- typo corrected

Reviewer 2 Report

This manuscript could be accepted to Vaccines. The topic is very modern and important. There are limited data concerning the immunogenicity and reactogenicity of COVID-19 vaccines in children. A total of 110 children, 5-11 years old were vaccinated with two doses (with a 3-week interval between doses) of the Pfizer-BioNTech COVID-19 vaccine and were followed for 21, 90, and 180 days after vaccination for immunogenicity, adverse events and break through infections. Ninety days after the 1st vaccine dose, the GeoMean (CI 95%) of IgG ascended to 1291.0 BAU (929.6-1790.2) for uninfected children and 1670.0 BAU (1131.0-2466.0) for Infected children. 180 days after receiving the 1st dose of the vaccine, the titers decreased to 535.5 BAU (288.4-993.6) for the uninfected children, while only a small decline was detected among infected children - 1479.0 (878.2-2490.0). The neutralizing antibodies titer almost did not change over time in the uninfected children, and even elevated for the infected children. Of the 110 vaccinated children, 75.5% were infected, with only mild COVID-19 infection symptoms. Children vaccination was found safe, with mild, mostly local, and of short duration reported AE's. No serious adverse events (SAEs) were reported after vaccination. The durability of 2 doses of vaccine in children is longer, thus a booster may not be needed as early as in adults. The introduction provide sufficient background. The research methodology is adequate and modern. The results are clearly presented. The amount of data is large. The conclusions supported by the data. The manuscript good illustrated and interesting to read. English language and style are fine, and may be very minor polishing from native speaker is recommended. Also, some more details perspectives regarding the future research could be formulated in conclusions section. Overall, this nice manuscript could be accepted for publication.

Author Response

We thank you for your prompt and constructive review. Below are the answers relating to all the comments:

Comments and Suggestions for Authors

This manuscript could be accepted to Vaccines. The topic is very modern and important. There are limited data concerning the immunogenicity and reactogenicity of COVID-19 vaccines in children. A total of 110 children, 5-11 years old were vaccinated with two doses (with a 3-week interval between doses) of the Pfizer-BioNTech COVID-19 vaccine and were followed for 21, 90, and 180 days after vaccination for immunogenicity, adverse events and break through infections. Ninety days after the 1st vaccine dose, the GeoMean (CI 95%) of IgG ascended to 1291.0 BAU (929.6-1790.2) for uninfected children and 1670.0 BAU (1131.0-2466.0) for Infected children. 180 days after receiving the 1st dose of the vaccine, the titers decreased to 535.5 BAU (288.4-993.6) for the uninfected children, while only a small decline was detected among infected children - 1479.0 (878.2-2490.0). The neutralizing antibodies titer almost did not change over time in the uninfected children, and even elevated for the infected children. Of the 110 vaccinated children, 75.5% were infected, with only mild COVID-19 infection symptoms. Children vaccination was found safe, with mild, mostly local, and of short duration reported AE's. No serious adverse events (SAEs) were reported after vaccination. The durability of 2 doses of vaccine in children is longer, thus a booster may not be needed as early as in adults. The introduction provides sufficient background. The research methodology is adequate and modern. The results are clearly presented. The amount of data is large. The conclusions supported by the data. The manuscript good illustrated and interesting to read. English language and style are fine, and may be very minor polishing from native speaker is recommended. Also, some more details perspectives regarding the future research could be formulated in conclusions section. Overall, this nice manuscript could be accepted for publication.

Answer- Thank you very much for your professional and warm review. The manuscript has been proof read again, and some small correction were made.  A sentence in the conclusion was added concerning perspectives regarding future research(line 300):"  Additionaly, future research is needed to determine the incidence of severe infections in healthy children vaccinated with two doses in order to define the need for a booster dose and adaptation for VOC's in such populations."

Reviewer 3 Report

In this manuscript, Joseph et al., analyzed the immunogenicity and reactogenicity of BNT162b2 in children. They found that although the vaccination induced good immune response, there were no serious problem after vaccination. Overall, the experiments and analysis are well performed. I have a few comments for this manuscript.

Minor

1.     Figure 1A and 1B: The difference of IgG and neutralizing titer between infected and uninfected donors were significant? Please add the statics information in the Figure or in the Result.

2.     Figure 1B: The authors performed neutralizing assay. Did they used virus pseudotyped by Omicron-spike or Wuhan spike? Ideally, they should use Omicron-pseudotyped virus. If not, they should at least clarify this in the Material and Method.

3.     Do the authors have any information if the donors have any underlying disease or not.

Author Response

We thank you for your prompt and constructive review. Below are the answers relating to all the comments

Comments and Suggestions for Authors

In this manuscript, Joseph et al., analyzed the immunogenicity and reactogenicity of BNT162b2 in children. They found that although the vaccination induced good immune response, there were no serious problem after vaccination. Overall, the experiments and analysis are well performed. I have a few comments for this manuscript.

Minor

  1. Figure 1A and 1B: The difference of IgG and neutralizing titer between infected and uninfected donors were significant? Please add the statics information in the Figure or in the Result.

Answer –We added the significance in the text of the results (line 180-190) and in table S4 and S5: ´… children and 1670.0 BAU (1131.0-2466.0) amongst the infected children (p>0.05).  The GMT (CI 95%) of IgG antibodies 180 days after receiving the 1st vaccine dose decreased to 535.5 BAU (288.4-993.6) in the uninfected children, while only a small decline was detected among infected children - 1479.0 BAU (878.2-2490.0) (*p<0.05).

Figure 1B (and Table S4) shows the neutralizing antibodies titers during the study. At baseline (day 0), the GMT (CI 95%) was a titer of 0.10 (0.10-0.12). At day 21 the titer increased to 78.8 (54.4-114.3). By 90 days after receiving the first vaccine dose, neutralizing antibody GMT titers increased to 664.0 (425.6-1036.0) in the uninfected group and 1057.0 (645.9-1730.0) in the infected group (p>0.05). At the end of study, 180 days after receiving the first vaccine dose, the neutralizing antibody GMT titer did not wane and the GMT in the uninfected group was 658.8 (291.9-1487.0). Among the infected group, a continued increase in neutralizing antibody titers was observed to 2250.0 (1185.0-4272.0) (*p<0.05

  1. Figure 1B: The authors performed neutralizing assay. Did they used virus pseudotyped by Omicron-spike or Wuhan spike? Ideally, they should use Omicron-pseudotyped virus. If not, they should at least clarify this in the Material and Method.

Answer:  The ps-virus used was of the Wuhan spike. Following the reviewer's comment, we added this information to the Materials and Methods section (line 139).

  1. Do the authors have any information if the donors have any underlying disease or not.

Answer – Thank you for the comment, a sentence detailing this issue was added in the first paragraph of the results: " According to parental reports, except for one child who has hyperactive airway disease treated with inhalers, none of the children included in our cohorts had chronic illness. Four of the children receive regular treatment for attention deficit hyperactivity disorder (ADHD). "